# Factors Associated with Post-Transplant Active Epstein-Barr Virus Infection and Lymphoproliferative Disease in Hematopoietic Stem Cell Transplant Recipients: A Systematic Review and Meta-Analysis

**DOI:** 10.3390/vaccines9030288

**Published:** 2021-03-19

**Authors:** Pascal Roland Enok Bonong, Monica Zahreddine, Chantal Buteau, Michel Duval, Louise Laporte, Jacques Lacroix, Caroline Alfieri, Helen Trottier

**Affiliations:** 1Department of Social and Preventive Medicine, Université de Montréal, CHU Sainte-Justine, Montréal, QC H3T 1C5, Canada; pascal.roland.enok.bonong@umontreal.ca (P.R.E.B.); monica.zahreddine@umontreal.ca (M.Z.); 2Department of Pediatrics, Division of Infectious Diseases, CHU Sainte-Justine, Université de Montréal, Montréal, QC H3T 1C5, Canada; chantal.buteau.hsj@ssss.gouv.qc.ca; 3Department of Pediatrics, Division of Hematology-Oncology, CHU Sainte-Justine, Université de Montréal, Montréal, QC H3T 1C5, Canada; michel.duval@umontreal.ca; 4Research Center of CHU Sainte-Justine, Montréal, QC H3T 1C5, Canada; louise.laporte.hsj@ssss.gouv.qc.ca; 5Department of Pediatrics, Division of Pediatric Intensive Care Medicine, CHU Sainte-Justine, Université de Montréal, Montréal, QC H3T 1C5, Canada; jacques.lacroix.med@ssss.gouv.qc.ca; 6Departement of Microbiology, Infectiology and Immunology, Université de Montréal, CHU Sainte-Justine Research Center, Montréal, QC H3T 1C5, Canada; carolina.alfieri@umontreal.ca

**Keywords:** Epstein–Barr virus (EBV), human herpesvirus-4 (HHV-4), risk factors, post-transplant lymphoproliferative disease (PTLD), EBV reactivation, hematopoietic stem cell transplant (HSCT)

## Abstract

This systematic review was undertaken to identify risk factors associated with post-transplant Epstein–Barr virus (EBV) active infection and post-transplant lymphoproliferative disease (PTLD) in pediatric and adult recipients of hematopoietic stem cell transplants (HSCT). A literature search was conducted in PubMed and EMBASE to identify studies published until 30 June 2020. Descriptive information was extracted for each individual study, and data were compiled for individual risk factors, including, when possible, relative risks with 95% confidence intervals and/or *p*-values. Meta-analyses were planned when possible. The methodological quality and potential for bias of included studies were also evaluated. Of the 3362 titles retrieved, 77 were included (62 for EBV infection and 22 for PTLD). The overall quality of the studies was strong. Several risk factors were explored in these studies, but few statistically significant associations were identified. The use of anti-thymocyte globulin (ATG) was identified as the most important risk factor positively associated with post-transplant active EBV infection and with PTLD. The pooled relative risks obtained using the random-effect model were 5.26 (95% CI: 2.92–9.45) and 4.17 (95% CI: 2.61–6.68) for the association between ATG and post-transplant EBV infection and PTLD, respectively. Other risk factors for EBV and PTLD were found in the included studies, such as graft-versus-host disease, type of conditioning regimen or type of donor, but results are conflicting. In conclusion, the results of this systematic review indicate that ATG increases the risk of EBV infection and PTLD, but the link with all other factors is either nonexistent or much less convincing.

## 1. Introduction

Hematopoietic stem cell transplant (HSCT) recipients are at risk of developing post-transplant lymphoproliferative disease (PTLD) following primary or reactivated infection by the Epstein–Barr virus (EBV) [1,2,3,4,5,6,7]. EBV is a ubiquitous human herpesvirus with a seroprevalence approximating 50–55% of the pediatric population living in countries with high hygienic standards and reaching 90–99% by mid-adulthood [8,9,10]. EBV is the etiologic agent of infectious mononucleosis and is also associated with the development of some cancers, most notably Hodgkin’s lymphoma, African Burkitt’s lymphoma and nasopharyngeal carcinoma [11,12,13], as well as lymphoproliferative disease in immunocompromised individuals [14]. After primary infection, EBV establishes latent infection in B cells [15]. In immunocompetent individuals, primary infection is often subclinical and latent infection is usually well controlled by the immune system throughout life. However, when the cytolytic T-lymphocyte arm of the immune system is suppressed, primary infection can be more consequential, and latent EBV can reactivate to cause a spectrum of EBV-associated diseases ranging from fever, EBV end-organ diseases, such as pneumonia, hepatitis and encephalitis, to PTLD [16]. PTLD is a complex disorder whereby an interplay of factors is involved in facilitating tumorigenesis [17].

The occurrence of PTLD in patients receiving an allogeneic HSCT can reach 24%, depending on the presence of risk factors [16,18,19]. The highest incidence of PTLD is seen in the first six months post-transplant, with most cases occurring during the first year post-transplant [4,6,20]. Infants are generally at higher risk because they are most often EBV-naïve before transplant [21]. Several clinical risk factors have been associated with EBV infection and PTLD in HSCT, including T-cell depletion of the graft, use of unrelated donors or of two or more HLA-mismatches in related donors, use of anti-lymphocyte serum for prevention or treatment of acute graft-versus-host disease (GvHD) and use of anti-CD3 monoclonal antibodies for acute GvHD [19].

The rapid increase of EBV viral load (EBV-VL) in the blood is a well-documented predictive biomarker of EBV-associated diseases. Following transplantation, regular monitoring of EBV-VL is usually performed for better management of patients who show large spikes in VL [22,23]. A reduction in the intensity of immunosuppression or treatment with the anti-CD20 monoclonal (rituximab) is effective in decreasing EBV-VL to prevent PTLD [24]. In patients receiving HSCT, rituximab use is the more common option [7]. However, both methods for lowering EBV-VL have important disadvantages. Reduction in the intensity of immunosuppression can increase the risk of GvHD [6], while rituximab use in patients who are already immunosuppressed can incur the development of other fatal infections [6]. Rituximab targets CD20-expressing malignant B cells as well as all mature B cells, thus impeding the production of antimicrobial immunoglobulins [6]. There is clearly an important clinical advantage in preventing EBV disease rather than attempting to cure it. Numerous studies have sought to better understand the determinants of EBV infection following allogeneic HSCT. The literature contains numerous important studies that consider one or more risk factors in small to large sample sizes of patients with different characteristics; however, no systematic review is available summarizing the determinants of EBV infection in HSCT. Therefore, the aim of this work was to synthesize, through a systematic review and meta-analysis, the risk factors associated with active EBV infection and with PTLD in HSCT recipients.

## 2. Methods

We conducted, using Medline and EMBASE, a systematic search of all articles on risk factors for active EBV infection (including EBV primary infection as well as EBV reactivation) and PTLD in pediatric and adult recipients of HSCT published in peer-reviewed journals between 1946 and 30 June 2020. A non-exhaustive list of concepts and keywords was obtained by referring to articles related to active EBV infection and PTLD; the list was broadened using medical subject heading (MeSH) descriptors in Medline and Emtree in EMBASE. The ovidSP interface was used to search in both databases. The search equations are presented in Appendix A. The selection of the articles was done in four steps: (1) title exploration, (2) review of abstracts, (3) review of the articles’ contents, and (4) review of the references of selected articles. All selection steps were performed independently by two authors (PE, MZ); in cases of disagreement, a third author (HT) was solicited for a consensual decision.

For the systematic review, three inclusion conditions were applied: (1) the study population had to be composed of pediatric and/or adult HSCT recipients; (2) risk factors for EBV infection or for PTLD had to be analyzed using univariate and/or multivariate statistical methods; and (3) the paper had to be in English or French. Abstracts, conference papers, congress papers, editorials, guidelines, reviews and case reports were excluded from the systematic review.

Two independent authors (PE and MZ) extracted the following information from the selected articles: authors, publication year, location, study type, post-transplant follow-up duration, transplant type, sample size, population (child or adult, and median or mean age, range or interquartile range), the definition of PTLD or definition of EBV infection, frequency of EBV-VL testing, blood compartment used to measure EBV-VL, and statistical methods used. In addition, for all potential risk factors explored in the studies, point estimates, such as odds ratio (OR), risk ratio (RR), the hazard ratio (HR) and subhazard ratio (SHR), confidence intervals (CI) and *p*-values were extracted when reported. In some cases, the corresponding author was contacted to clarify ambiguities.

The quality of each individual study was independently evaluated by two authors (PE & MZ) using a modified version of the Effective Public Health Practice Project (EPHPP) quality assessment tool for quantitative studies [25,26]. The quality assessment was based on the following components: selection bias, study design, confounders and data collection methods; it was rated as strong, moderate or weak (from high-quality to low-quality) according to the definition presented in Appendix A.

Finally, risk factors explored in these studies were described by providing the total number of studies showing a statistically significant association contrasted to the total number of studies investigating the risk factor.

The data reported made it possible to perform a meta-analysis solely to measure the association between the use of anti-thymocyte globulin (ATG) and two outcomes: post-transplant EBV infection and PTLD. Studies using multivariate analysis were considered for the meta-analysis except for the study by Liu et al. [27] because only the *p*-value was reported (not the measure of association). Since post-transplant EBV infection is not a rare event in this population, to obtain pooled estimates, results from studies that reported adjusted HR or SHR were combined separately from those that reported adjusted OR. This distinction was not made for PTLD, which is a relatively less frequent event. Adjusted estimates were combined using the inverse variance method with the fixed-effect model or random-effect model. The choice between these two models was guided by the value of statistic I, which revealed the proportion of the total variance observed due to a real difference in the measures of effects between studies. The fixed-effect model was used when I^2^ < 25% and the random-effect model when I^2^ ≥ 25% [28,29]. We also performed a sensitivity analysis to assess the contribution of each study to the pooled estimate. To this end, the pooled estimate was recalculated, each time excluding only one of the studies considered [30]. The analysis was performed with software R version 3.6.1.

## 3. Results

In total, 3362 titles were identified in the research bases, 1883 in EMBASE and 1479 in Medline. Once duplicates and papers with exclusion criteria were removed, 77 articles [4,16,23,24,27,31,32,33,34,35,36,37,38,39,40,41,42,43,44,45,46,47,48,49,50,51,52,53,54,55,56,57,58,59,60,61,62,63,64,65,66,67,68,69,70,71,72,73,74,75,76,77,78,79,80,81,82,83,84,85,86,87,88,89,90,91,92,93,94,95,96,97,98,99,100,101,102] fulfilled the inclusion criteria for our systematic review (62 for EBV and 22 for PTLD). Among the 22 articles selected for PTLD, seven were also retained for EBV. Detailed information on the selection procedure is provided in the flow diagram (Figure 1). Among the 62 articles included to analyze risk factors for post-transplant EBV infection, two relate exactly to the same patient cohort (Bogunia-Kubik et al. [36] and Bogunia-Kubik et al. [35]) and 11 relate to non-disjoint samples (include some of the same patients) (Cesaro et al. [41] and Cesaro et al. [42]; Liu et al. [74] and Liu et al. [27]; Xuan et al. [99] and Liu et al. [74]; Liu et al. [75] and Liu et al. [73]; Wang et al. [97] and Ru et al. [86]; Zhou et al. [102] and Zhou et al. [101]). However, none of these studies but one [101] were excluded from the qualitative synthesis for duplication because the risk factors explored were different. The study by Zhou et al. [102] was excluded because all variables in this paper were explored using univariate analysis and were considered in the study by Zhou et al. [101] using multivariate analysis. With respect to the analysis of PTLD risk factors, the studies by Sundin et al. [103] and Omar et al. [81] were discarded because Uhlin et al. [94] explored the same factors and sample population as these two studies. Hoegh-Petersen et al. [60] and Kalra et al. [67] used non-disjoint samples. However, these two studies were retained in the review because the risk factors explored were not completely identical. For the same risk factors explored in both studies, those from Kalra et al. [67] were retained as the analyses were done on a larger sample. In addition, the Ali et al. [31] and Althubaiti et al. [32] studies were carried out with non-disjoint samples, but both were retained because different variables were explored.

Characteristics of the selected studies are described in Table 1; more details are provided in Appendix A. Briefly, among the 77 studies, seven were conducted in France, seven in Italy, six in Poland, one in Belgium, three in Spain, six in the United Kingdom, one in Finland, five in the United States, three in Japan, 16 in China, two in Korea, one in Russia, five in Canada, three in Sweden, one in Turkey, three in Germany, two in the Netherlands, one in Greece, one in Portugal and three were multi-national. Twenty-three studies were prospective, 51 were retrospective, two were case–control studies and one was a randomized control trial. The sample size ranged from 26 to 64,539 HSCT recipients (Figure 2). It is noteworthy that most studies were performed with pooled pediatric and adult populations (n = 41), while 19 included only children and 17 only adults.

The definition of post-transplant EBV infection and the diagnostic criteria for PTLD differed among studies. Active post-transplant EBV infection was diagnosed when the EBV-VL in blood, determined using a PCR test, was above a given threshold. In some cases, thresholds were not readily comparable because there was no direct conversion between the units of measurement used. In two studies [45,91], active EBV infection was defined as a reactivation event because all patients showed positive EBV serology when the follow-up period started. In other studies, no distinction was made between primary infection and reactivation: both were considered active EBV infection. There was also some variability between studies with respect to the frequency of PCR testing, but it was performed weekly in most studies during the early post-transplant period. The type of specimen tested by PCR varied, with peripheral blood in 19 studies, plasma in 11 studies, serum in five studies, serum or plasma in one study, whole blood in 13 studies, peripheral blood or whole blood in two studies, and whole blood and plasma in one study; 10 studies provided no information on specimen type. The method used to diagnose PTLD was not detailed in one study. The length of follow-up was an important source of variation between studies; in some cases the follow-up period was not reported [31,32,34,40,46,48,52,55,59,61,63,73,81,83,86,90,94,98,100].

Various statistical methods were used. Logistic regression was used in 13 studies, Cox model in 22, survival analysis using the log-rank test in one, multiple linear regression in two, Fine and Gray competitive risk model in 10 and Poisson regression for grouped survival data in one. Among the 28 studies employing univariate analysis, the statistical method was not explicitly reported in one study; one study used univariate logistic regression, another used the univariate Cox model, another used time-dependent landmark, while other studies used at least one of the following tests: Log-rank test, Gray’s test, Chi 2 test, Wilcoxon nonparametric test, Kruskal–Wallis test, Fisher’s exact test, Mann–Whitney test or Wald test. Among the 49 studies in which a multivariate analysis was performed, the criteria for selecting variables for the multivariate model were explicitly indicated in 20. Variables with a *p*-value ˂ 0.1 in univariate analysis were retained for multivariate analysis in eight studies, a *p*-value ˂ 0.2 in three, a *p*-value ˂ 0.3 in one, a *p*-value ˂ 0.05 in one; a *p*-value < 0.05 was used for the multivariate analysis in three studies. In four studies, the investigators used a *p*-value < 0.1 in the univariate analysis combined with a *p*-value < 0.05 in multivariate analysis. Altogether, 52 studies were considered as properly adjusted for confounding bias.

Appendix A reports the quality assessment of the 74 articles included in the review according to the outcome. Regarding post-transplant EBV infection, 27 (42.9%) were rated “strong”, 15 (23.8%) “moderate” and 21 (33.3%) “weak”. For PTLD, 12 (57.1%) were rated “strong”, three (14.3%) “moderate” and six (28.6%) “weak”. The lack of information on retention and potential for selection bias, as well as the absence of control for potential confounding bias, were the main contributors to the lower overall rating of articles. It is important to note that the absence of control for confusion in several articles could be justified by the fact that their main focus did not involve analysis of factors associated with either post-transplant EBV infection or PTLD.

Appendix A provides a detailed list of the risk factors for post-transplant EBV infection and for PTLD explored in the 77 included studies with a description, when possible, of the measures of association and CI and/or *p*-value. Figure 3 presents, for every individual risk factor, the total number of studies that investigated the risk factor contrasted to the number that showed a statistically significant association. Table 2 presents a summary of the risk factors (with measures of association) for post-transplant EBV infection and for PTLD that were analyzed in studies using multivariate analysis. The presence of GvHD, use of ATG and type of conditioning regimen were the three risk factors most frequently associated with EBV infection and PTLD.

### 3.1. Graft-versus-Host Disease

The association between acute (a)GvHD and post-transplant EBV infection was examined in 21 studies [4,39,41,46,51,52,56,57,59,65,66,68,74,76,81,85,86,88,89,91,101]. Six statistically significant associations were highlighted using different grade categorization of the outcome. In two studies, aGvHD was dichotomized in grade ≥3 versus <3; one of them (Juvonen et al. [66]) showed that patients with aGvHD grade ≥3 had a higher risk of active EBV infection (HR = 1.70 (95% CI: 1.11–2.62)). However, this result was not corroborated by other studies [51,91,101]. Among eight studies [4,39,41,57,59,74,86,89] that compared grade ≥2 versus <2, two [59,89] showed that an aGvHD grade ≥2 significantly increased the risk of active EBV infection, one reported a positive association without reporting the relative risk (Hiwarkar et al. [58]), and the other (Sirvent-Von Bueltzingsloewen et al. [89]) reported an OR = 3.4 (95% CI: 1.2–9.7). The potential effect of aGvHD, however, was not confirmed by the other studies [4,39,41,57,74]. One study (Peric et al. [85]) categorized aGvHD according to grade 0–1, grade 2 and grades 3–4 and did not show a statistically significant association with active EBV infection. Seven studies considered the presence versus absence of aGvHD; three highlighted a statistically significant association. Elmahdi et al. [52] showed that the presence of aGvHD, whatever the grade, increased the risk of EBV infection (HR = 3.29 (95% CI: 1.26–8.58)). Similar results were obtained by Cohen et al. [46] (OR = 2.2 (95% CI: 2.12–15.08)) and by Omar et al. [81] who showed that patients with aGvHD had on average a higher EBV-VL than patients without aGvHD (*p* = 0.009). However, these results were not corroborated in four other studies [56,65,68,76]. Seven studies examined if chronic (c) GvHD was a risk factor of post-transplant EBV infection [4,41,46,68,74,86,88], two of which showed a statistically significant relationship [68,86]. Two studies [24,69] did not differentiate aGvHD and cGvHD; one [69] showed a statistically significant association.

In regards to PTLD, its occurrence was associated with aGvHD grade ≥2 in Landgren et al. (RR = 1.7 [1.2–2.5)) [70], Uhlin et al. (SHR = 2.65 (1.32–5.35)) [94] and Fujimoto et al. (HR = 1.93 (1.48–2.52)) [55]. No statistically significant association was identified in eight other studies [16,27,46,56,57,88,95,98] that explored the association between aGvHD and PTLD. Four studies [16,27,70,88] analyzed the association between cGvHD and PTLD; only Landgren et al. [70] found a statistically significant association (RR = 2.0 [1.1–3.2)). In contrast, the study by Kalra et al. [67]. showed a lower risk of PTLD in patients with aGvHD grade ≥2 or cGvHD that required systemic therapy (SHR = 0.47; *p* = 0.04).

### 3.2. Graft-versus-Host Disease Prophylaxis/Treatment

ATG use appears to be an important risk factor for the development of active post-transplant EBV infection or PTLD. Among 15 multivariate studies [4,41,45,46,51,53,56,66,68,69,74,79,85,86,96] that examined the association between ATG and active post-transplant EBV infection, 10 found a statistically significant association: Cesaro et al. [41] (HR = 13.0 (95% CI: 2–96)), Fan et al. [53] (OR = 7.69 (95% CI: 1.17–50.49)), Juvonen et al. [66] (HR = 5.78 (95% CI: 2.47–13.5)), Liu et al. [74] (HR = 14.081 (95% CI: 6.02–32.92)), Peric et al. [85] (SHR = 4.9 (95% CI: 1.1–21.0)), Van Esser et al. [96] (HR = 3.4 (95% CI: 1.6–1)), Gao et al. [56] (HR = 6.3 (95% CI: 1.6–24.0)), Düver et al. [51] (OR = 10.68 (95% CI: 1.15–98.86)), Ru et al. [86] (HR = 4.29 (95% CI: 2.64–6.97)) and Kullberg-Lindh et al. [68] (slope = 1.34; *p* = 0.004). All studies compared patients who received ATG versus those who did not, but one: Van Esser et al. [96] reported the risk of EBV infection in patients receiving T-cell depleted (TCD) grafts with ATG versus patients receiving non-TCD grafts. A statistically significant association between active post-transplant EBV infection and TCD grafts was shown by Bordon et al. [37] (*p* = 0.04) as well as for CD4^+^ depleted grafts by Torre-Cisneros et al. [91] (OR = 11.5 (95% CI: 5.8–22.8)). Corticosteroid use for GvHD prophylaxis (OR = 23.68 (95% CI: 1.92–291.45)) was associated with EBV infection in the study by Fan et al. [53].

An association between ATG and PTLD was reported by Landgren et al. [70] (RR = 3.8 [2.5–5.8)), Van der Velden et al. [95] (OR = 2.4 (1.3–4.2)), Liu et al. [27] (*p* = 0.038), Xuan et al. [16] (HR = 13.03 (1.67–101.58)) as well by Fujimoto et al. [55] (HR = 6.13 (95% CI: 4.33–8.68] for GvHD prophylaxis and HR = 2.09 (95% CI: 1.17–3.72] for GvHD treatment). The association was not statistically significant in the study by Gao et al. [56] (HR = 2.9 (95% CI: 0.3–27.5)). Brunstein et al. [38] found a higher risk for the composite outcome ‘post-transplant EBV infection or PTLD’ in patients with non-myeloablative conditioning regimen (NMAC) + ATG (HR = 15.4 (2.0–116.1)), but a similar risk for those receiving NMAC without ATG (HR = 0.7 (0.1–6.5)) compared to those who received myeloablative conditioning (MAC). This highlights the role of ATG as a significant risk factor. Buyck et al. [40] reported a dose-response relationship: the risk of PTLD increased with the number of prior courses of ATG (HR = 7.23 (1.67–31.32)). Lin et al. [72] found a higher risk of post-transplant EBV infection in patients, who received a higher dose of ATG (10.0 mg/kg versus 7.5 mg/kg: HR = 2.02 (95% CI: 1.37–2.97)). The study by Cohen et al. [46] compared patients who received Campath versus ATG; no statistically significant association was found (unadjusted OR = 0.56 (0.15–2.05)).

The meta-analyses that we performed are presented in Figure 4 and Figure 5. The pooled HR for the association between ATG use and post-transplant EBV infection obtained using the random-effect model was 5.26 (95% CI: 2.92–9.45) with an I^2^ = 63.2% (Figure 4). We performed sensitivity analyses by recalculating the pooled estimate after excluding only one study at a time: the results vary between 4.13 and 6.49, and the I^2^ heterogeneity statistic varies between 22% and 69%. The studies by Laberko et al. [69] and Liu et al. [74] had the greatest influence on the pooled estimate and on the level of heterogeneity. However, regardless of the study excluded, the overall result remains statistically significant. With respect to studies that estimated an adjusted OR to report the association between ATG and post-transplant EBV infection, the pooled estimate was 2.74 [1.03–7.31] and I^2^ = 40.3% (Figure 5). The sensitivity analyses highlighted a variation of the pooled estimate from 2.07 to 4.00 and of I^2^ from 28% to 58%. The studies by Christopeit et al. [45] and Cohen et al. [46] had the greatest influence on the pooled estimate and heterogeneity. The pooled estimate was no longer significant if a single study was removed from the analysis, except for the study by Christopeit et al. [45], which was carried out with the smallest sample. The pooled RR for the association between ATG and PTLD obtained using the random-effect model was 4.17 (95% CI: 2.61–6.68) with an I^2^ = 56.7%. The sensitivity analysis revealed that the pooled estimate ranged from 3.34 to 5.02 and the I^2^ from 9% to 67%. The studies by Fujimoto et al. [55] and Van der Velden et al. [95] had the biggest influence on the pooled estimate and the I^2^. The sensitivity analysis did not question the statistically significant association between ATG and PTLD.

The results of these meta-analyses should be understood cautiously given the high-level of heterogeneity observed between studies. Due to the small number of articles, we did not explore the sources of heterogeneity further by performing a subgroup analysis or a meta-regression.

### 3.3. Other Risk Factors

Other possible risk factors were analyzed in the retained studies; these factors were not associated with EBV infection or PTLD, or their relationship was more ambiguous (Table 2 and Appendix A). The association between the primary diagnosis and post-transplant active EBV infection was explored in several reports [4,23,35,39,46,56,59,69,85,86,88,101]; three showed a strong positive and statistically significant association [23,39,88]. According to Carpenter et al. [23], the risk of active EBV infection was greater in patients with Hodgkin’s lymphoma (HR = 3.53 (95% CI: 1.51–8.25)) or chronic lymphocytic leukemia (HR = 3.77 (95% CI: 1.38–10.32)) compared to patients with acute myeloid leukemia. Sanz et al. [88] reported that the risk was greater in patients with Hodgkin disease compared to other patients (SHR = 11.6 (95% CI: 3.4–40.0)). However, Burns et al. [39] found that the risk of active post-transplant EBV infection was lower in patients with non-Hodgkin’s lymphoma compared to patients with acute myeloid leukemia/myelodysplastic syndrome (AML/MDS) (HR = 0.18 (95% CI: 0.05–0.57)). Furthermore, Fujimoto et al. [55] showed a higher risk of PTLD in patients with aplastic anemia compared to those with AML/MDS (HR = 5.19 (95% CI: 3.32–8.11)). No statistically significant association between the PTLD outcome and the patient’s primary diagnosis was found in the other studies [46,56,88,94,95,98].

The use of reduced-intensity conditioning regimen was determined as a risk factor for post-transplant EBV infection by Sanz et al. [88] (SHR = 6.0 (2.0–17.6)) and as a risk factor for PTLD by Sanz et al. [88] (SHR = 5.5 (1.8–17.1)) and Uhlin et al. [94] (SHR = 3.25 (1.53–6.89)). Two studies reported that intensified myeloablative conditioning regimen (MAC) increased the risk of post-transplant EBV infection: Liu et al. [74] (HR = 1.72 (1.03–2.88)) and Lin et al. [72] (HR = 1.73 (95% CI: 1.18–2.54)). Liu et al. [74] also found an association between intensified MAC and PTLD (*p* = 0.018), but Xuan et al. [16] found an association in the opposite direction (standard versus intensified regimen: HR = 4.46 (1.20–16.61)). Gao et al. [56] found that the use of fludarabine would increase the risk of PTLD (HR = 3.8 (1.4–10.6)). EBV viral load was higher in the study by Kullberg-Lindh et al. [68] when total body irradiation (TBI) was performed.

Recipient age did not seem to be an important risk factor for active EBV infection post-transplant. Of the 20 studies [4,23,35,39,50,51,52,56,65,68,69,74,76,85,86,88,89,93,96,101] to consider this factor, only Bogunia-Kubik et al. [35] highlighted a statistically significant association, showing that the propensity for active post-transplant EBV infection was higher in people over 25 years compared to others (OR = 1.54 (95% CI: 1.14–2.70)). Concerning the association between age and PTLD, only two [56,70] of the 10 studies [16,27,40,56,67,70,88,94,95,98] that explored this factor indicated a statistically significant association. A higher risk of PTLD has been observed in patients aged 50 years or more (RR = 5.1 (2.8–8.7)) [70]. Conversely, a lower risk of PTLD was observed in patients 40 years or older in another study (HR = 0.4 (0.2–0.9)) [56].

Several studies used multivariate analysis to examine the relationship between recipient sex and post-transplant active EBV infection [4,35,39,50,52,56,65,68,69,72,74,76,85,86,88,89,96,101] or PTLD [16,27,40,56,88,94,95,98]; none found a significant association. Three out of six studies [35,53,56,65,79,101] that analyzed the association between donor sex and post-HSCT active EBV infection showed a statistically significant association but in the opposite direction. In two studies, the risk for active EBV infection post-HSCT was higher in patients receiving a male donor transplant [53,56] while, in the other, patients receiving a female donor transplant appeared to be at greater risk [65]. The only study [56] that explored the association between donor sex and PTLD did not find a statistically significant association. Moreover, no statistically significant association was found between the donor/recipient sex combination and post-transplant EBV infection [35,41,53,57,65]. Among all studies that examined the sex of the dyad donor/recipient and PTLD [57,82,94,98], only one [82] found a statistically significant association suggesting a higher risk of PTLD in patients who received a transplant from a different sex donor.

With regard to viral infections, a study by Zallio et al. [24] suggests that the risk of active post-transplant EBV infection is higher in patients with CMV reactivation compared to those who are CMV negative (*p* < 0.05). Similar results were found by Gao et al. (HR = 5.9 (2.5–13.9)) [56] and by Zhou et al. (HR = 97.75 (9.48–1008.30)) [101]. Xu et al. [98] and Gao et al. [56] found that the risk of PTLD was higher in patients with CMV DNAemia: HR = 5.68 (1.17–7.57) and HR = 11.6 (1.2–114.4), respectively. According to Lakerko et al. [69], the risk of post-transplant EBV active infection was higher among EBV seronegative patients (compared to seropositive patients) in HSCT recipients receiving a graft from an EBV-seropositive donor (HR = 2.85 (95% CI: 1.12–7.28)). Lin et al. [72] found a higher risk of post-transplant EBV infection in EBV-seropositive patients who received an EBV-negative graft (HR = 1.58 (1.01–2.46)). Other risk factors associated with active EBV infection post-transplant include: (1) two human genotypes, namely the interferon-ɣ (IFNɣ) gene 3/3 (OR = 7.28; *p* = 0.005) [36] and the CC-chemokine receptor-5 (CCR5) (OR = 0.17 (95% CI: 0.03–0.80)) [35], (2) the volume of platelets transfused (>2530 vs. <1260 mL) (HR = 2.19 (95% CI: 1.21–3.97)) [92], (3) unrelated or mismatched related donor (*p* = 0.04) [81], (4) unrelated donor ((HR = 8.8, *p* = 0.030) [76], (OR = 5.05 (1.24–20.63)) [51], (HR = 2.63 (1.02–6.67)) [93]), (5) HLA incompatibility ((OR = 5 [1.5–16.4)) [89], (HR = 1.83 (1.27–2.63)) [86]), (6) CD3^+^ count in the graft ≥ median (OR = 0.11 (0.02–0.78)) [4], (7) CD3^+^CD8^+^ count in the graft ≥ median (OR = 0.05 (0.01–0.43)) [45], (8) CD34^+^ count in the graft >1.35 × 10^6^/kg (HR = 2.6 (1.5–4.6)) [96], (9) CD4^+^ lymphocyte/µl at one month after HSCT ≥ 50 (OR = 0.1 (0.02–0.481)) [4], (10) Vδ2^+^ T cell count 30 days post-transplant (HR = 0.347 (0.161–0.747)) [73], (11) IgM level ≥ median 30 days after HSCT (HR = 0.27 (0.10–0.75)) [98], (12) proportion (%) of NKp30/total NK cells one month after HSCT (HR = 0.96 (0.918–0.998)) [100], and (13) prior HSCT (HR = 2.6 (1.1–6.4)) [57]. Also, a higher risk of post-transplant EBV infection was observed by Tsoumakas et al. [93] in patients receiving a peripheral blood transplant compared to those receiving a bone marrow transplant (HR = 2.51 (1.04–6.05)), but an opposite result was found by Wang et al. [97] (HR = 18.69; *p* < 0.001).

Other factors associated with PTLD include: (1) CD8^+^ count (≥median vs. <median) 30 days after HSCT (HR = 0.34 (0.13–0.92)), (2) prior HSCT ((HR = 2.6 (1.1–6.4)) [57], (RR = 3.5 [1.7–6.3)) [70]), (3) splenectomy (SHR = 4.81 (1.51–15.4)) [94], (4) infusion of mesenchymal stromal cells (SHR = 3.05 (1.25–7.48)) [94], (5) a stepwise increase of EBV-DNA by 1 log (HR = 2.9 (1.7–4.8)) [96], (6) HLA DRB1*11:01 (SHR = 4.85 (1.57–14.97)) [82], and (7) HLA mismatch (SHR = 5.89 (2.43–14.3)) [94]. Fujimoto et al. [55] found that, compared to matched related donor grafts, the risk of PTLD is higher when using mismatched related donor grafts (HR = 4.39 (2.39–8.07)), matched unrelated donor grafts (HR = 4.08 (2.39–6.99)), mismatched unrelated donor grafts (HR = 3.20 (1.58–6.47)) or cord blood grafts (HR = 8.03 (4.72–13.7)).

## 4. Discussion

This systematic review includes 77 papers. It aims to characterize risk factors associated with active post-transplant EBV infection and PTLD in HSCT recipients. Active EBV infection can result in rapidly increasing EBV-VL, which is a high-risk marker for PTLD development. Proper identification of the risk factors associated with active EBV infection and PTLD is needed for effective patient management.

In this systematic review, we focused on risk factors explored in published studies; very few statistically significant associations were found. The use of ATG was identified as one of the most important risk factors for the development of active post-transplant EBV infection and PTLD. The pooled relative risks estimated from the meta-analysis that was carried out confirmed a positive and statistically significant association between ATG and EBV infection (RR = 3.98 (95% CI: 2.20–7.18) and PTLD (RR = 3.69 (95% CI: 2.24–6.08)). ATG is a potent immunosuppressive agent that obliterates the T-cell pool [104,105,106], thereby enabling reactivation of latent EBV contained in mature B cells along with the malignant expansion of infected cells [104]. In the HSCT setting, ATG is used for the prevention of aGvHD, given its ability to target and deplete T lymphocytes [107,108].

Some studies included in this review also found an association with the presence of GvHD, which is an immune-mediated complication of HSCT whereby donor T cells present in the graft initiate an alloreactive process that ultimately causes destruction of host tissues [109]. aGvHD usually occurs within the first three months post-transplant and is categorized into four grades ranging from 1 (light disease) to 4 (severe disease) [110]. cGvHD usually occurs beyond the initial three months post-transplant. The pathophysiology of GvHD, especially that of cGvHD, is complex [111]. T and B lymphocytes are probably involved in the pathophysiology of GvHD, although the mechanism linking these cells to GvHD is not well-known [108]. In short, the etiology of GvHD is complex, and it is difficult to conclude whether GvHD is an independent risk factor for EBV and PTLD or whether the relationship found in some studies is the result of confounding by indication related to the use of ATG. The analyses that we are currently running among pediatric HSCT recipients recruited in our TREASuRE cohort study [112] confirm that EBV is strongly associated with ATG but not with GvHD, following adequate control for confounding bias.

Many other variables were analyzed in the 77 included studies, but results were either inconsistent, failed to find an association, or limited in terms of the number of studies that investigated the risk factor. Some studies showed that primary diagnosis was associated with post-transplant active EBV infection [23,39,88], more specifically in the case of Hodgkin disease [23,88]. Some forms of Hodgkin’s lymphoma are etiologically linked to EBV [113,114] and may occur in individuals who are not able to properly control EBV infection. These individuals may be thought to be more susceptible to other EBV diseases (such as post-transplant active EBV infection) along the continuum of care, but HSCT should have corrected any immune cell problem. Although interesting, further studies are needed to confirm the potential association between Hodgkin’s disease and post-transplant active EBV infection in HSCT patients.

Discordant results were found for other variables, and, in other cases, the number of studies investigating risk factors was limited. These variables are recipient age, recipient gender, donor type, conditioning regimen, graft source, graft history, graft content (CD34^+^, CD3^+^, CD8^+^, CD3^+^/CD8^+^), genotype (IFNɣ gene 3/3, CCR5), splenectomy, mesenchymal stromal cells, donor gender and transfusion (red blood cells, platelets, plasma) (Appendix A). In our recent study, although no relationship was statistically found between EBV and blood product transfusion, we linked a case of EBV infection in an EBV-seronegative pediatric HSCT recipient to a blood donor through viral genotype analysis [112]. One cause of discordant results is the heterogeneity observed among the various studies, most notably with regard to the different specimen types used to perform PCR tests (Appendix A). The sensitivity of PCR tests is greater when whole blood is used as opposed to plasma [115]. Other sources of discordance include variations in the statistical approach and experimental design. We also noted the absence of controls for confounding and failure to report results when associations lacked statistical significance. In addition, only 42.9% of studies included in the systematic review of factors associated with post-transplant EBV infection were classified as being of strong quality, and 23.8% were classified as moderate quality; with respect to PTLD as an outcome, the proportions were, respectively 57.1% and 14.3%. An important risk for bias includes uncontrolled confounding bias and the lack of information on retention, a potential source of selection bias in cohort studies.

This review was not able to discern whether differences exist between children and adults. While statistical power was higher in studies combining both groups, differences in terms of risk factors may exist. Immune restoration through T-cell reconstitution after transplantation is different in children and adults [116], and risk factors may differ. It should be noted that 25 of the 77 studies selected in this systematic review have a sample size of less than 100; therefore, it is possible that type II error may explain why positive associations were not statistically significant in many studies. Moreover, the included studies were limited to the identification of factors associated with the first occurrence of active EBV infection post-transplant, although during follow-up a patient may experience several episodes of active EBV infection [23,116]. This latter aspect should be considered in order to better understand the dynamics of the evolution of active EBV infection post-transplant in HSCT recipients. Risk factors for the occurrence of active EBV infection may be different from those that explain the dynamics of infection. Finally, there was insufficient information on attrition, which may be the primary source of selection bias in this type of study. While we initially intended to perform a meta-analysis of all risk factors associated with active EBV infection and PTLD, this was not possible because of the diversity of outcome definitions, the variability in the definition of risk factors and the non-systematic reporting of point estimates, confidence intervals and *p*-values. However, as indicated above, a meta-analysis was carried out to measure the association between ATG use and post-transplant EBV infection and PTLD, respectively. The results, however, must be considered with caution, as the definition of outcome was quite variable from one study to another. Based on all the above arguments, further studies using large cohorts of children and adults are needed to better elucidate the determinants of active EBV infection and PTLD among HSCT recipients.

In conclusion, we found ATG as the most important risk factor for the development of active post-transplant EBV infection and PTLD in HSCT patients. ATG considerably increases the risk of EBV and PTLD. Other risk factors have been linked with EBV and PTLD in studies, such as GvHD or type of donor, but the association for these other factors is less clear due to conflicting results, the potential for bias, particularly confounding, or because of the low number of studies that considered these risk factors. Further studies using large cohorts of children and adults with appropriate control for confounding are needed to better characterize other determinants of active EBV infection and PTLD among HSCT recipients.

## Figures and Tables

**Figure 1 vaccines-09-00288-f001:**
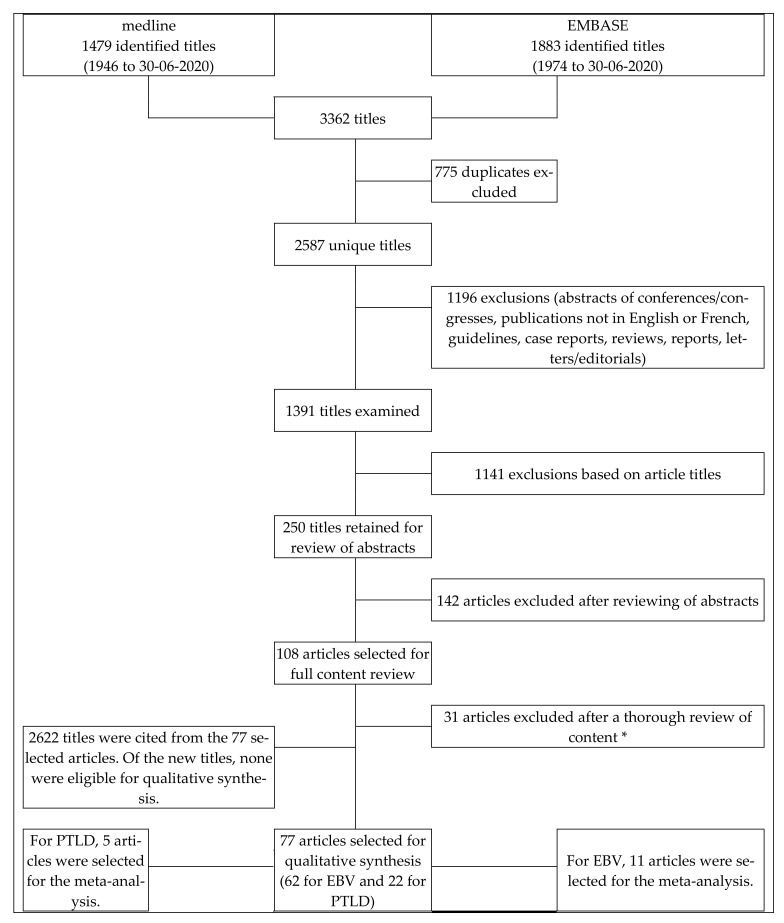
Search strategy flowchart. * The reasons for exclusion of these articles were as follows: Two articles were excluded because their sample is a subset of the sample from two other articles. There was no univariate or multivariate statistical analysis for the identification of risk factors for post-transplant active EBV infection or PTLD in 26 articles and in three articles. EBV post-transplant infection was combined with other viral infections in a single variable.

**Figure 2 vaccines-09-00288-f002:**
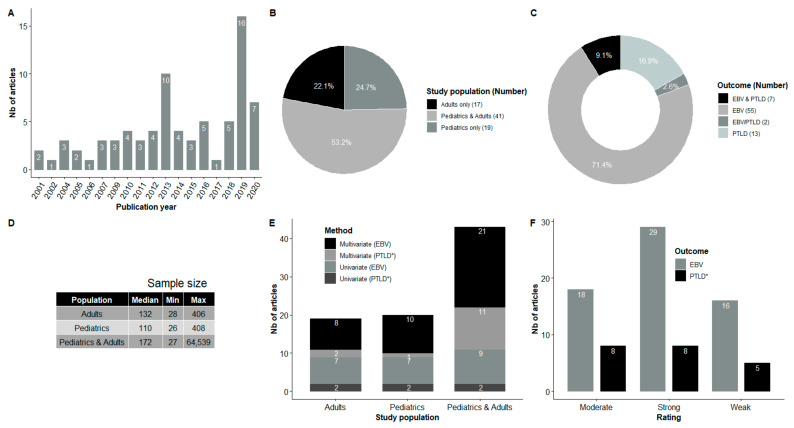
Summary of some characteristics of the studies included in the systematic review. (**A**) Number of studies by year of publication; (**B**) Proportion (number) of studies by type of population; (**C**) Proportion (number) of studies according to the type of outcome; (**D**) Descriptive statistics on sample size by type of population; (**E**) Number of studies according to the type of statistical analysis carried out, the type of population and the outcome; (**F**) Number of studies by type of outcome and by quality level. EBV and PTLD: The two outcomes were studied separately in the same article; EBV/PTLD: The two outcomes were combined into one. * Studies with the outcome PTLD/EBV and the studies with outcome PTLD were considered together.

**Figure 3 vaccines-09-00288-f003:**
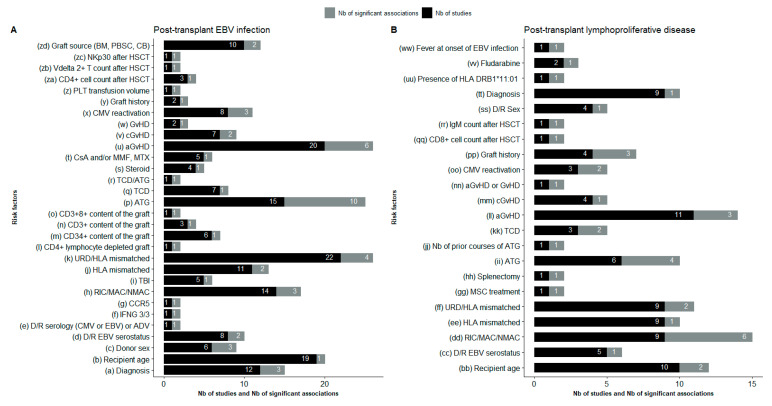
Summary of risk factors for post-transplant EBV infection (**A**) and for PTLD (**B**) explored in the studies that controlled for confounding. Abbreviations: ADV: adenovirus; aGvHD: acute graft-versus-host disease; ATG: anti-thymocyte globulin; BM: bone marrow; CB: cord blood; CCR5: C–C chemokine receptor 5; cGvHD: chronic graft-versus-host disease; CMV: cytomegalovirus; CsA: cyclosporine A; D/R: donor/recipient; GvHD: graft-versus-host disease; HLA: human leukocyte antigen; HSCT: hematopoietic stem cell transplantation; IFNG: interferon-γ gene; MAC: myeloablative conditioning; MMF: mycophenolate mofetil; MSC: mesenchymal stromal cells; MTX: methotrexate; NK: natural killer cells; NMAC: nonmyeloablative conditioning; PBSC: peripheral blood stem cells; PLT: platelets; RIC: reduced-intensity conditioning; TBI: total body irradiation; TCD: T-cell depletion; URD: unrelated donor.

**Figure 4 vaccines-09-00288-f004:**
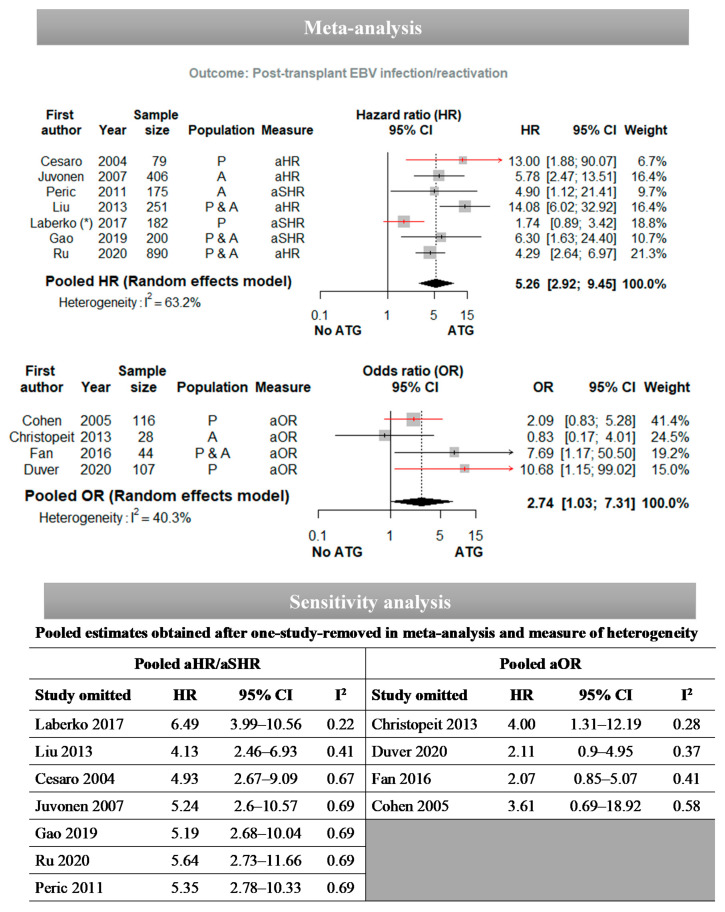
Forest plots for the association between ATG use and post-transplant EBV infection according to studies estimating adjusted HR/SHR and adjusted OR. (*) In the study by Laberko et al., two estimates of the hazard ratio (HR) of the association between the use of ATG and post-transplant EBV infection were reported, corresponding to the use of horse ATG on one hand and rabbit ATG on the other. These two HRs were combined using a meta-analysis with inverse variance as a method. The results obtained were used to carry out the meta-analysis, including the other studies. Abbreviations: OR: odds ratio; HR: hazard ratio; SHR: subhazard ratio; CI: confidence intervals; ATG: anti-thymocyte globulin.

**Figure 5 vaccines-09-00288-f005:**
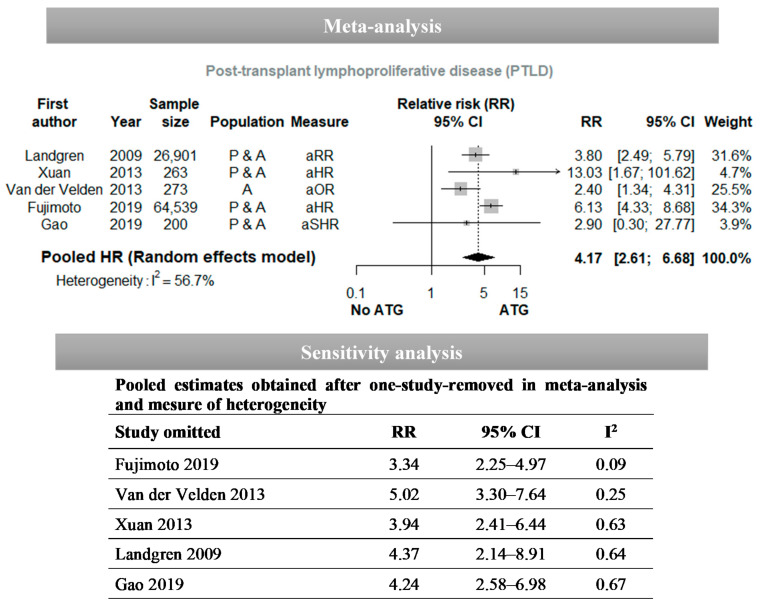
Forest plots for the association between ATG use and post-transplant lymphoproliferative disease (PTLD). Abbreviations: OR: odds ratio; HR: hazard ratio; SHR: subhazard ratio; CI: confidence intervals; ATG: anti-thymocyte globulin.

**Table 1 vaccines-09-00288-t001:** Characteristics of the 77 studies included in the systematic review.

First Author, Year	Country	Study Type	Study Population	Sample Size	Outcome	Median (Range) of Follow-Up	Statistical Analysis	Overall Rating (Appendix A)
Ali, 2019 [31]	Canada	Retrospective	P	408	PTLD	NR	Univariate	Weak
Althubaiti, 2019 [32]	Canada	Retrospective	P	26	PTLD	NR	Univariate	Weak
Atay, 2018 [33]	Turkey	Retrospective	P	171	EBV ^╧^	14 months	Univariate	Weak
Auger, 2014 [34]	France	Retrospective	A	190	EBV	36.6 months (95% IC 31.5–45.7)	Multivariate	Weak
Bogunia-Kubik, 2007 [35]	Poland	Retrospective	P and A	92	EBV	NR	Multivariate	Strong
Bogunia-Kubik, 2005 [36]	Poland	Retrospective	P and A	83	EBV	NR	Multivariate	Strong
Bordon, 2012 [37]	Belgium	Retrospective	P	80	EBV	NR	Multivariate	Moderate
Brunstein, 2006 [38]	USA	Multicenter retrospective	P and A	335	EBV/PTLD	1.2 (77 days–9.2 years)	Multivariate	Moderate
Burns, 2016 [39]	United Kingdom	Retrospective	P and A	186	EBV	28 months	Multivariate	Strong
Buyck, 2009 [40]	United Kingdom	Retrospective	P and A	87	PTLD	NR	Multivariate	Moderate
Carpenter, 2010 [23]	United Kingdom	Retrospective	P and A	111 ^a^	EBV	2.4 years	Multivariate	Strong
Cesaro, 2004 [41]	Italy	Retrospective	P	79 ^b^	EBV	NR	Multivariate	Moderate
Cesaro, 2010 [42]	Italy	Retrospective	P	89	EBV	NR	Univariate	Weak
Chiereghin, 2016 [43]	Italy	Prospective	P	28	EBV	7.1 months	Univariate	Weak
Chiereghin, 2019 [44]	Italy	Prospective	P and A	51	EBV	NR	Univariate	Weak
Christopeit, 2013 [45]	USA	Retrospective	A	28 ^c^	EBV	NR	Multivariate	Moderate
Cohen, 2005 [46]	United Kingdom	Prospective	P	128	EBV	NR	Multivariate	Moderate
Cohen, 2005 [46]	United Kingdom	Prospective	P	128	PTLD	NR	Multivariate	Moderate
Comoli, 2007 [47]	Italy	Prospective	P and A	27	EBV	23 months	Univariate	Weak
Czyżewski, 2019 [48]	Poland	Retrospective multicenter study	P and A	1569	EBV	NR	Univariate	Weak
D’Aveni, 2011 [49]	France	Retrospective	P and A	40 ^d^	EBV	NR	Univariate	Weak
Dumas, 2013 [50]	France	Multicenter retrospective	P and A	175	EBV	NR	Multivariate	Moderate
Düver, 2020 [51]	Germany	Retrospective	P	107	EBV	365 (range: 22–365) days	Multivariate	Strong
Elmahdi, 2016 [52]	Japan	Retrospective	P	37	EBV	NR	Multivariate	Moderate
Fan, 2016 [53]	China	Retrospective	P and A	44 ^e^	EBV ^╧^	NR	Multivariate	Moderate
Figgins, 2019 [54]	USA	Retrospective	A	123	EBV	12.8 (range: 1.0–23.1) months	Univariate	Weak
Fujimoto, 2019 [55]	Japan	Multicenter retrospective	P and A	64,539	PTLD	NR	Multivariate	Strong
Gao, 2019 [56]	China	Retrospective	P and A	200	EBV	NR	Multivariate	Strong
Gao, 2019 [56]	China	Retrospective	P and A	200	PTLD	NR	Multivariate	Strong
Garcia-Cadenas, 2015 [57]	Spain	Prospective	A	93	EBV	NR	Multivariate	Strong
Garcia-Cadenas, 2015 [57]	Spain	Prospective	A	93	PTLD	NR	Multivariate	Strong
Han, 2014 [58]	Korea	Retrospective	P	248	EBV	NR	Univariate	Weak
Hiwarkar, 2013 [59]	United Kingdom	Retrospective	P	278	EBV	NR	Multivariate	Moderate
Hoegh-Petersen, 2011 [60]	Canada	Retrospective	A	307	PTLD	375 (28–1727) days	Univariate	Weak
Hoshino, 2001 [61]	Japan	Prospective	P and A	38	EBV	NR	Univariate	Weak
Islam, 2010 [62]	United Kingdom	Retrospective	P and A	83	EBV	4.2 (0.9–8.1) years	Univariate	Weak
Issa, 2019 [63]	USA	Retrospective	A	357	EBV	NR	Univariate	Weak
Kutnik, 2019 [64]	Poland	Retrospective	P	198	EBV	12 months	Univariate	Weak
Jaskula, 2010 [65]	Poland	Prospective	P and A	102	EBV	NR	Multivariate	Moderate
Juvonen, 2007 [66]	Finland	Retrospective	A	406	EBV	NR	Multivariate	Strong
Kalra, 2018 [67]	Canada	Retrospective	P and A	554	PTLD	509 days	Multivariate	Strong
Kullberg-Lindh, 2015 [68]	Sweden	Retrospective	P	47	EBV	NR	Multivariate	Strong
Laberko, 2017 [69]	Russia	Retrospective	P	182	EBV	27 months	Multivariate	Strong
Landgren, 2009 [70]	CIBMTR	Multicenter retrospective	P and A	26,901	PTLD	>12 months	Multivariate	Strong
Li, 2018 [71]	China	Retrospective	P	62	EBV ^╧^	32.5 (0.5–132) months	Univariate	Weak
Lin, 2019 [72]	China	Multicenter randomized study	P and A	408	EBV	NR	Multivariate	Strong
Liu, 2020 [73]	China	Prospective	A	170	EBV	NR	Multivariate	Strong
Liu, 2020 [73]	China	Prospective	A	170	PTLD	NR	Univariate	Weak
Liu, 2013 [74]	China	Prospective	P and A	251 ^f^	EBV	327 (27–1408) days	Multivariate	Strong
Liu, 2013 [27]	China	Prospective	P and A	172	EBV	495 (45–1158) days	Multivariate	Strong
Liu, 2013 [27]	China	Prospective	P and A	172	PTLD	495 (45–1158) days	Multivariate	Strong
Liu, 2018 [75]	China	Prospective	A	132	EBV	NR	Univariate ^‡^	Strong
Marinho-Dias, 2019 [76]	Portugal	Prospective	P and A	40	EBV	>120 days	Multivariate	Strong
Meijer, 2004 [77]	Netherlands	Prospective	A	78 ^g^	EBV	(6–32) months	Univariate	Weak
Mountjoy, 2020 [78]	USA	Retrospective	A	209	EBV	Non-ATG group 677 (7–3147) days ATG group 504 (33–2156) days	Univariate	Weak
Neumann, 2018 [79]	Germany	Case–control	A	44	EBV ^╧^	NR	Univariate ^§^	Strong
Nowak, 2019 [80]	Poland	Retrospective	P and A	239	EBV	2.1 (0.2–67.8) months	Univariate	Weak
Omar, 2009 [81]	Sweden	Prospective	P and A	131	EBV	NR	Multivariate	Moderate
Pagliuca, 2019 [82]	France	Retrospective	P and A	208	PTLD	47.33 (3.18–126.20) months	Multivariate	Strong
Park, 2020 [83]	Korea	Retrospective	P and A	114	EBV	NR	Univariate	Weak
Patriarca, 2013 [4]	Italy	Prospective	A	100 ^h^	EBV	7 (2–36) months	Multivariate	Strong
Peric, 2012 [84]	France	Retrospective	A	33	EBV	468 (92–1277) days	Univariate	Weak
Peric, 2011 [85]	France	Retrospective	A	175	EBV	655 (92–1542) days	Multivariate	Strong
Ru, 2020 [86]	China	Retrospective	P and A	890	EBV	NR	Multivariate	Strong
Rustia, 2016 [87]	USA	Retrospective	P	140	EBV	NR	Univariate	Weak
Sanz, 2014 [88]	Spain	Retrospective	P and A	288	EBV	>6 months	Multivariate	Strong
Sanz, 2014 [88]	Spain	Retrospective	P and A	288	PTLD	>6 months	Multivariate	Strong
Sirvent-von Bueltzingsloewen, 2002 [89]	France	Multicenter prospective	P and A	85 ^i^	EBV	306 (26–867) days	Multivariate	Strong
Styczynski, 2013 [90]	EBMT	Multicenter retrospective	P and A	4466	PTLD	NR	Univariate	Weak
Torre-Cisneros, 2004 [91]	Spain	Prospective	P and A	100 ^j^	EBV	NR	Multivariate	Moderate
Trottier, 2012 [92]	Canada	Retrospective	P	238	EBV	NR	Multivariate	Moderate
Tsoumakas, 2019 [93]	Greece	Prospective	P	110	EBV	≥1 year	Multivariate	Strong
Uhlin, 2014 [94]	Sweden	Retrospective	P and A	1021	PTLD	NR	Multivariate	Strong
Van der Velden, 2013 [95]	Netherlands	Retrospective	A	273	EBV/PTLD	≥6 months	Multivariate	Moderate
Van Esser, 2001 [96]	Italy, Germany, Netherlands	Multicenter prospective	P and A	152	EBV	NR	Multivariate	Strong
Van Esser, 2001 [96]	Italy, Germany, Netherlands	Multicenter prospective	P and A	152	PTLD	NR	Multivariate	Strong
Wang, 2019 [97]	China	Retrospective	P and A	186	EBV	NR	Multivariate	Strong
Xu, 2015 [98]	China	Case–control	P and A	180	PTLD	NR	Multivariate	Strong
Xuan, 2012 [99]	China	Prospective	P and A	185	EBV	319 (27–1194) days	Multivariate	Strong
Xuan, 2013 [16]	China	Prospective	P and A	263	PTLD	374 (27–1554) days	Multivariate	Strong
Yu, 2019 [100]	China	Prospective	P and A	90	EBV	NR	Multivariate	Moderate
Zallio, 2013 [24]	Italy	Prospective	A	100	EBV	NR	Multivariate	Moderate
Zhou, 2020 [101]	China	Retrospective	P and A	131	EBV	59.2 (range: 2.03–113.8) months	Multivariate	Strong
Zhou, 2020 [102]	China	Retrospective	P and A	160	PTLD	64.7 (range: 2.03–113.8) months	Univariate	Weak

^a^ Alemtuzumab was considered in the conditioning protocol of all patients, and only patients with at least 6 months of follow-up were considered. ^b^ Almost all patients received the standard conditioning regimen. ^c^ All of these patients had positive EBV serology, survived beyond 40 days and received cyclosporine beyond 30 days post-transplant. ^d^ Of the 40 patients, five were excluded: three because of related early transplant mortality and two dues to relapse before 60 days of follow-up. ^e^ All patients in the study had positive CMV serology and negative PCR tests for herpesviruses (EBV, CMV, and HHV-6) one week after transplantation. ^f^ All patients had a negative EBV PCR test at the start of follow-up. ^g^ All except 1 (receiving bone marrow) received a peripheral blood stem cell graft. ^h^ All patients had a follow-up duration > 30 days post-transplant. ^i^ Five patients with post-transplant lymphoproliferative syndrome were excluded. Analysis of risk factors for EBV reactivation involved 80 patients. ^j^ All patients had positive EBV serology before transplantation. ^§^ The individuals were matched according to the variables age, diagnosis, and conditioning regimen. ^‡^ Chi 2 test and Mann–Whitney U test were used to verify that the distributions of potential confounding factors were not significantly different. ^╧^ The outcome has not been explicitly defined. Abbreviation: ATG: anti-thymocyte globulin; CIBMTR: Center for International Blood and Marrow Transplant Research; EBV: Epstein–Barr virus; EMBT: European Group for Blood and Marrow Transplantation; NR: not reported; P: pediatrics; P and A: pediatrics and adults; PTLD: post-transplant lymphoproliferative disease.

**Table 2 vaccines-09-00288-t002:** Summary of risk factors for post-transplant EBV infection and for PTLD in the studies using multivariate analysis.

First Author, Year	Outcome	Study Population	Risk Factors		Estimate (95% CI); *p*-Value *	
			**Recipient age**			
Bogunia-Kubik, 2007 [35]	EBV	P and A	> vs. ≤25 years		**OR = 1.54 (1.136–2.703); *p* = 0.034**	
Ru, 2020 [86]	EBV	P and A	<30 vs. ≥30 years		HR = 1.041 (0.763–1.420); *p* = 0.799	
Düver, 2020 [51]	EBV	P	Age (continuous)		OR = 1.08 (1.00–1.17); *p* = 0.057	
Kullberg-Lindh, 2011 [68]	EBV	P	Continuous		Slope = −0.06; *p* = 0.09	
Gao, 2019 [56]	PTLD	P and A	≥40 vs. <40 years		**HR = 0.4 (0.2–0.9); *p* = 0.032**	
Landgren, 2009 [70]	PTLD	P and A	≥50 years		**RR = 5.1 (2.8–8.7)**	
			**Diagnosis**			
Burns, 2016 [39]	EBV	P and A	NHL vs. AML/MDS		**HR = 0.18 (0.05–0.57); *p* = 0.004**	
			ALL vs. AML/MDS		HR = 0.89 (0.45–1.75); *p* = 0.734	
			HL vs. AML/MDS		HR = 1.63 (0.64–4.16); *p* = 0.308	
			CLL vs. AML/MDS		HR = 0.87 (0.41–1.85); *p* = 0.724	
			MPD vs. AML/MDS		HR = 0.95 (0.43–2.11); *p* = 0.907	
			Other vs. AML/MDS		HR = 3.01 (0.94–9.65); *P* = 0.063	
Carpenter, 2010 [23]	EBV	P and A	HL vs. AML		**HR = 3.53 (1.51–8.25); *p* = 0.004**	
			NHL vs. AML		HR = 0.678 (0.249–1.848); *p* = 0.448	
			MPD vs. AML		HR = 2.01 (0.828–4.858); *p* = 0.123	
			CLL vs. AML		**HR = 3.767 (1.375–10.322); *p* = 0.01**	
			Other disease vs. AML		HR = 1.449 (0.486–4.319); *p* = 0.506	
Sanz, 2014 [88]	EBV	P and A	Hodgkin’s disease vs. other diagnosis		**SHR = 11.6 (3.4–40.0); *p* < 0.0001**	
Zhou, 2020 [101]	EBV	P and A	Underlying disease (AA vs. AL)		HR = 4.369 (0.484–39.451); *p* = 0.189	
Fujimoto, 2019 [55]	PTLD	P and A	ALL vs. AML/MDS		HR = 1.08 (0.75–1.57); *p* = 0.68	
			CML/MPD vs. AML/MDS		HR = 1.55 (0.89–2.69); *p* = 0.12	
			Lymphoid malignancies vs. AML/MDS		HR = 1.33 (0.92–1.92); *p* = 0.13	
			AA vs. AML/MDS		**HR = 5.19 (3.32–8.11); *p* < 0.001**	
			Others vs. AML/MDS		HR = 1.94 (0.97–3.89); *p* = 0.06	
			**Genotype**			
Bogunia-Kubik, 2005 [36]	EBV	P and A	Recipient having IFNG 3/3 genotype vs. other IFNG		**OR = 7.28; *p* = 0.005**	
Bogunia-Kubik, 2007 [35]	EBV	P and A	Presence of CCR5 deletion mutation (yes vs. no)		**OR = 0.17 (0.034–0.803); *p* = 0.026**	
Pagliuca, 2019 [82]	PTLD	P and A	Presence of HLA DRB1*11:01 (yes vs. no)		**SHR = 4.85 (1.57–14.97); *p* = 0.006**	
			**Recipient, donor EBV, CMV serostatus**			
Hiwarkar, 2013 [59]	EBV	P	D+ and R+ (CMV or EBV) or host adenoviral infection		**Significant, but NR**	
Laberko, 2017 [69]	EBV	P and A	EBV D+/R− vs. D+/R+		**HR = 2.85 (1.12–7.28); *p* = 0.028**	
			EBV D−/R+ vs. D+/R+		HR = 0.32 (0.05–2.0); *p* = 0.22	
			EBV D−/R− vs. D+/R+		No events	
			EBV Unknown vs. D+/R+		HR = 1.23 (0.53–2.9); *p* = 0.63	
Lin, 2019 [72]	EBV	P and A	D/R EBV serostatus (D−/R+ vs. Other)		**HR = 1.58 (1.01–2.46); *p* = 0.046**	
Uhlin, 2014 [94]	PTLD	P and A	EBV D+ R− vs. Other		**SHR = 4.97 (2.30–10.7); *p* < 0.001**	
Brunstein, 2006 [38]	EBV/PTLD	P and A	CMV (R− vs. R+)		HR = 3.0 (0.9–9.7) *p* = 0.07	
			**Donor sex**			
Fan, 2016 [53]	EBV	P and A	Male donor		**OR = 13.24 (2.006–87.387); *p* = 0.007**	
Jaskula, 2010 [65]	EBV	P and A	Female donor		**OR = 2.816; *p* = 0.044**	
			**Donor type**			
Düver, 2020 [51]	EBV	P	Unrelated donor vs. Related donor		**OR = 5.05 (1.24–20.63); *p* = 0.024**	
Marinho-Dias, 2019 [76]	EBV	P and A	Unrelated donor (yes vs. no)		**HR = 8.8, *p* = 0.030 at D + 150**	
Tsoumakas, 2019 [93]	EBV	P	Related donor vs. unrelated donor		**HR = 0.38 (0.15–0.98); *p* = 0.045**	
Omar, 2009 [81]	EBV	P and A	URD + MMRD vs. HLA-matched donor		***p* = 0.04**	
Pagliuca, 2019 [82]	PTLD	P and A	Unrelated (yes vs. no)		SHR = 2.11 (1.00–4.45); *p* = 0.051	
Fujimoto, 2019 [55]	PTLD	P and A	MMRD vs. MRD		**HR = 4.39 (2.39–8.07); *p* < 0.001**	
			MURD vs. MRD		**HR = 4.08 (2.39–6.99); *p* < 0.001**	
			MMURD vs. MRD		**HR = 3.20 (1.58–6.47); *p* = 0.001**	
			CB vs. MRD		**HR = 8.03 (4.72–13.7); *p* < 0.001**	
Sirvent-von Bueltzingsloewen, 2002 [89]	EBV	P and A	HLA incompatibility (yes vs. no)		**OR = 5 (1.5–16.4)**	
Torre-Cisneros, 2004 [91]	EBV	P and A	No HLA-matched sibling donor		HR = 2.1 (0.8–6.2); *p* = 0.069	
Gao, 2019 [56]	EBV	P and A	Haploidentical donors vs. matched sibling donors		HR = 2.0 (0.8–5.1); *p* = 0.130	
Ru, 2020 [86]	EBV	P and A	HLA-haploidentical vs. HLA-identical		**HR = 1.830 (1.275–2.627); *p* = 0.001**	
Gao, 2019 [56]	PTLD	P and A	Haploidentical donors vs. matched sibling donors		HR = 2.0 (0.5–8.3); *p* = 0.350	
Uhlin, 2014 [94]	PTLD	P and A	HLA mismatch vs. match		**SHR = 5.89 (2.43–14.3) *p* < 0.001**	
			**Graft source**			
Tsoumakas, 2019 [93]	EBV	P	PBSC vs. BM		**HR = 2.51 (1.04–6.05); *p* = 0.041**	
Wang, 2019 [97]	EBV	P and A	PB + BM vs. PB		**HR = 7.89; *p* = 0.003**	
			BM vs. PB		**HR = 18.69; *p* < 0.001**	
			**Graft content**			
Christopeit, 2013 [45]	EBV	A	CD3^+^ (≥ vs. < median)		**OR = 0.11 (0.02–0.78); *p* = 0.027**	
			CD3^+^CD8^+^ (≥ vs. < median)		**OR = 0.05 (0.006–0.431); *p* = 0.007**	
Van Esser, 2001 [96]	EBV	P and A	CD34^+^ (>1.35 × 10^6^/kg)		**HR = 2.6 (1.5–4.6); *p* = 0.001**	
			**Conditioning regimens and GvHD prophylaxis/treatment**			
Kullberg-Lindh, 2011 [68]	EBV	P	TBI (yes vs. no)		**Slope = 1.60; *p* = 0.001**	
Liu, 2013 [74]	EBV	P and A	Intensified MAC vs. standard MAC		**HR = 1.72 (1.03–2.88); *p* = 0.038**	
Lin, 2019 [72]	EBV	P and A	Intensified conditioning vs. standard MAC		**HR = 1.73 (1.18–2.54); *p* = 0.005**	
Sanz, 2014 [88]	EBV	P and A	RIC vs. MAC		**SHR = 6.0 (2.0–17.6); *p* = 0.001**	
	PTLD		RIC vs. MAC		**SHR = 5.5 (1.8–17.1); *p* = 0.003**	
Fujimoto, 2019 [55]	PTLD	P and A	RIC vs. MAC		HR = 0.82 (0.60–1.12); *p* = 0.22	
Uhlin, 2014 [94]	PTLD	P and A	RIC vs. no RIC		**SHR = 3.25 (1.53–6.89) *p* = 0.002**	
Xuan, 2013 [17]	PTLD	P and A	Standard vs. intensified		**HR = 4.46 (1.20–16.61); *p* = 0.026**	
Liu, 2013 [27]	PTLD	P and A	Intensified MAC vs. standard MAC		***p* = 0.018**	
Brunstein, 2006 [38]	EBV/PTLD	P and A	NMAC without ATG vs. MAC		HR = 0.7 (0.1–6.5); *p* = 0.51	
			NMAC with ATG vs. MAC		**HR = 15.4 (2.0–116.1); *p* < 0.01**	
Van der Velden, 2013 [95]	PTLD	A	MAC without ATG		**OR = 2.6 (1.05–7.15); *p* = 0.01**	
			NMAC with ATG		OR = 2.1 (0.92–4.8); *p* = 0.08	
Gao, 2019 [56]	PTLD	P and A	Use of fludarabine (yes vs. no)		**HR = 3.8 (1.4–10.6); *p* = 0.010**	
Cohen, 2005 [46]	EBV	P	ATG vs. Campath		OR = 2.09 (0.83–5.29)	
Cesaro, 2004 [41]	EBV	P	Use of ATG (yes vs. no)		**HR = 13.0 (2–96); *p* = 0.01**	
Düver, 2020 [51]	EBV	P	Use of ATG (yes vs. no)		**OR = 10.68 (1.15–98.86); *p* = 0.037**	
Gao, 2019 [56]	EBV	P and A	Use of ATG (yes vs. no)		**HR = 6.3 (1.6–24.0); *p* = 0.008**	
Kullberg-Lindh, 2011 [68]	EBV	P	Use of ATG (yes vs. no)		**Slope = 1.34; *p* = 0.004**	
Juvonen, 2007 [66]	EBV	A	Use of ATG (yes vs. no) ^╪^		**HR = 5.78 (2.47–13.5); *p* < 0.001**	
Peric, 2011 [85]	EBV	A	Use of ATG (yes vs. no)		**SHR = 4.9 (1.1–21.0); *p* = 0.03**	
Fan, 2016 [53]	EBV	P and A	Use of ATG (yes vs. no)		**OR = 7.69 (1.17–50.49); *p* = 0.034**	
Laberko, 2017 [69]	EBV	P and A	Horse ATG vs. no serotherapy		HR = 2.47 (0.95–6.38); *p* = 0.063	
			Rabbit ATG vs. no serotherapy		HR = 1.22 (0.467–3.18); *p* = 0.69	
Christopeit, 2013 [45]	EBV	A	Use of ATG (yes vs. no)		OR = 0.83 (0.17–4.01); *p* = 0.820	
Liu, 2013 [74]	EBV	P and A	Use of ATG (yes vs. no)		**HR = 14.08 (6.02–32.92); *p* < 0.001**	
Ru, 2020 [86]	EBV	P and A	Use of ATG (yes vs. no)		**HR = 4.288(2.638–6.97); *p* < 0.001**	
Liu, 2013 [27]	PTLD	P and A	Use of ATG (yes vs. no)		***p* = 0.038**	
Van der Velden, 2013 [95]	PTLD	A	Use of ATG (yes vs. no)		**OR = 2.4 (1.3–4.2) *p* = 0.001**	
Landgren, 2009 [70]	PTLD	P and A	Use of ATG (yes vs. no) ^╪^		**RR = 3.8 (2.5–5.8)**	
Xuan, 2013 [16]	PTLD	P and A	Use of ATG (yes vs. no)		**HR = 13.03 (1.67–101.58) *p* = 0.014**	
Fujimoto, 2019 [55]	PTLD	P and A	Use of ATG in conditioning regimen (yes vs. no)		**HR = 6.13 (4.33–8.68); *p* < 0.001**	
Fujimoto, 2019 [55]	PTLD	P and A	Use of ATG for GvHD treatment (yes vs. no) ^╪^		**HR = 2.09 (1.17–3.72); *p* = 0.01**	
Gao, 2019 [56]	PTLD	P and A	Use of ATG (yes vs. no)		HR = 2.9 (0.3–27.5); *p* = 0.350	
Lin, 2019 [72]	EBV	P and A	ATG dose (10.0 mg/kg vs. 7.5 mg/kg)		**HR = 2.02 (1.37–2.97); *p* < 0.001**	
Buyck, 2009 [40]	PTLD	P and A	Number of prior courses of ATG		**HR = 7.23 (1.67–31.32); *p* = 0.008;**	
Fan, 2016 [53]	EBV	P and A	MMF + CsA + prednisone vs. MMF + CsA		**OR = 23.68 (1.924–291.449); *p* = 0.013**	
Christopeit, 2013 [45]	EBV	A	CsA AUC (≥ vs. <6000 ng/mL x days)		**OR = 6.067 (1.107–33.238); *p* = 0.038**	
			**T-cell depletion**			
Bordon, 2012 [37]	EBV	P	In vivo TCD (yes vs. no)		***p* = 0.04**	
Torre-Cisneros, 2004 [91]	EBV	P and A	Use of CD4^+^ lymphocyte-depleted graft (yes vs. no)		**HR = 11.5 (5.8–22.8); *p* < 0.0001**	
Van Esser, 2001 [96]	EBV	P and A	TCD without ATG vs. non-TCD		HR = 1.5 (0.8–2.9); *p* = 0.3	
			TCD with ATG vs. non-TCD		**HR = 3.4 (1.6–7.1); *p* = 0.001**	
Landgren, 2009 [70]	PTLD	P and A	Broad lymphocyte depletion vs. no TCD		**RR = 3.1 (1.2–6.7)**	
			Selective TCD vs. no TCD		**RR = 9.4 (6.0–14.7)**	
			**Method of T-cell depletion**			
Landgren, 2009 [70]	PTLD	P and A	Alemtuzumab MoAb vs. no TCD		RR = 3.1 (0.7–8.4)	
			Elutriation/density gradient centrifugation vs. no TCD		RR = 3.2 (0.8–8.8)	
			Anti-T or anti-T + NK MoAb vs. no TCD		**RR = 8.4 (5.1–13)**	
			SRBC rosetting vs. no TCD		**RR = 14.6 (5.9–31)**	
			Lectins with/without SRBC or anti-T MoAb vs. no TCD		**RR = 15.8 (7.2–32)**	
			Unclassified/unknown method vs. no TCD		RR = 6.0 (0.96–20)	
			**Graft-versus-host disease**			
Cohen, 2005 [46]	EBV	P	aGvHD (yes vs. no)		**OR = 2.20 (2.12–15.08)**	
Elmahdi, 2016 [52]	EBV	P	aGvHD (yes vs. no)		**HR = 3.29 (1.26–8.58); *p* = 0.015**	
Hiwarkar, 2013 [59]	EBV	P	aGvHD ≥ grade II		**Significant, but NR**	
Kullberg-Lindh, 2011 [68]	EBV	P	cGvHD (yes vs. no)		**Slope = −1.12; *p* = 0.023**	
Juvonen, 2007 [66]	EBV	A	aGvHD ≥ grade III^╪^		**HR = 1.70 (1.11–2.62); *p* = 0.015**	
Sirvent-von Bueltzingsloewen, 2002 [89]	EBV	P and A	aGvHD ≥ grade II		**OR = 3.4 (1.2–9.7)**	
Omar, 2009 [81]	EBV	P and A	aGvHD (yes vs. no)		*p* = 0.009	
Gao, 2019 [56]	EBV	P and A	aGvHD (yes vs. no)		HR = 1.0 (0.7–1.6); *p* = 0.960	
Gao, 2019 [56]	PTLD	P and A	aGvHD (yes vs. no)		HR = 1.4 (0.5–3.8); *p* = 0.480	
Laberko, 2017 [69]	EBV	P and A	GvHD (yes vs. no)		**HR = 1.97 (1.04–3.72); *p* = 0.037**	
Landgren, 2009 [70]	PTLD	P and A	aGvHD ≥ grade II ^╪^		**RR = 1.7 (1.2–2.5)**	
Ru, 2020 [86]	EBV	P and A	aGvHD (grade II-IV vs. none or grade I)		HR = 1.26 (0.89–1.78); *p* = 0.193	
Fujimoto, 2019 [55]	PTLD	P and A	aGvHD grade II-IV (yes vs. no) ^╪^		**HR = 1.93 (1.48–2.52); *p* < 0.001**	
Uhlin, 2014 [94]	PTLD	P and A	aGvHD ≥ grade II		**SHR = 2.65 (1.32–5.35) *p* = 0.006**	
Landgren, 2009 [70]	PTLD	P and A	cGvHD moderate/severe or clinical extensive ^╪^		**RR = 2.0 (1.1–3.2)**	
Ru, 2020 [86]	EBV	P and A	cGvHD (yes vs. no)		**HR = 1.413 (1.013–1.971); *p* = 0.042**	
Kalra, 2018 [67]	PTLD	P and A	aGvHD grade II-IV or chronic NST (yes vs. no)		**SHR = 0.47, *p* = 0.04**	
			**Immunological reconstitution**			
Patriarca, 2013 [4]	EBV	A	Peripheral blood CD4^+^ lymphocyte/µL at +1 month after HSCT (≥50 vs. <50)		**OR = 0.1 (0.02–0.48); *p* = 0.004**	
Yu, 2019 [100]	EBV	P and A	NKp30 in 1-month post-transplant (1 M) (% of total NK cells)		**HR = 0.957 (0.918–0.998); *p* = 0.04**	
Liu, 2020 [73]	EBV	A	Vδ2^+^ cell recovery at day 30 post-transplantation		**HR = 0.347 (0.161–0.747); *p* = 0.007**	
Liu, 2020 [73]	EBV	A	CD8^+^ cell recovery at day 30 post-transplantation		HR = 0.499 (0.207–1.201); *p* = 0.121	
Xu, 2015 [98]	PTLD	P and A	CD8^+^ cell count at day 30 after HSCT (≥median vs. < median)		**HR = 0.34 (0.13–0.92) *p* = 0.033**	
	PTLD	P and A	IgM count at day 30 after HSCT (≥median vs. <median)		**HR = 0.27 (0.10–0.75) *p* = 0.012**	
			**CMV reactivation**			
Gao, 2019 [56]	EBV	P and A	CMV DNAemia (yes vs. no)		**HR = 5.9 (2.5–13.9); *p* < 0.001**	
Torre-Cisneros, 2004 [91]	EBV	P and A	CMV load > 2500 copies/mL		HR = 2.1 (0.9–7); *p* = 0.061	
Zallio, 2013 [24]	EBV	A	yes vs. no		Significant, but NR	
Zhou, 2020 [101]	EBV	P and A	CMV DNAemia (yes vs. no)		**HR = 97.754 (9.477–1008.304)**	
Gao, 2019 [56]	PTLD	P and A	CMV DNAemia (yes vs. no)		**HR = 11.6 (1.2–114.4); *p* = 0.036**	
Xu, 2015 [98]	PTLD	P and A	CMV DNAemia (yes vs. no)		**HR = 5.68 (1.17–27.57) *p* = 0.031**	
			**Transfusion**			
Trottier, 2012 [92]	EBV	P	RBC transfusion volume (mL)	<850 vs. 0	HR = 1.99 (0.47–8.44)	*p*-value trend = 0.047
				850–1890 vs. 0	HR = 2.40 (0.56–10.24)	
				>1890 vs. 0	HR = 2.86 (0.68–12.11)	
		P	FFP transfusion volume (mL)	≤200 vs. 0	HR = 0.70 (0.22–2.25)	*p*-value trend = 0.079
				>200 vs. 0	HR = 3.16 (1.00–11.17)	
		P	PLT transfusion volume (mL)	1260–2530 vs. <1260	HR = 1.65 (0.86–3.18)	*p*-value trend = 0.012
				>2530 vs. <1260	**HR = 2.19 (1.21–3.97)**	
			**Other factors**			
Garcia-Cadenas, 2015 [57]	EBV	A	Prior SCT (yes vs. no)		**HR: 2.6 (1.1–6.4); *p* = 0.04**	
	PTLD	A	Prior SCT (yes vs. no)		**HR: 6.4 (1.3–31.9); *p* = 0.02**	
Fujimoto, 2019 [55]	PTLD	P and A	Number of allogeneic HSCT (two or more vs. one)		**HR = 1.50 (1.05–2.15); *p* = 0.03**	
Landgren, 2009 [70]	PTLD	P and A	Second transplant (yes vs. no) ^╪^		**RR = 3.5 (1.7–6.3)**	
Uhlin, 2014 [94]	PTLD	P and A	Splenectomy (yes vs. no)		**SHR = 4.81 (1.51–15.4) *p* = 0.008**	
	PTLD	P and A	MSC treatment (yes vs. no)		**SHR = 3.05 (1.25–7.48) *p* = 0.015**	
Landgren, 2009 [70]	PTLD	P and A	2+ HLA MMRD or URD, no ATG, no selective TCD vs. matched sibling or 1 HLA-Ag mismatched relative		RR = 0.9 (0.3–2.2)	
			2+ HLA MMRD or URD, ATG and/or selective TCD vs. matched sibling or 1 HLA-Ag mismatched relative		**RR = 3.8 (2.4–6.1)**	
Van Esser, 2001 [96]	PTLD	P and A	A stepwise increase of EBV-DNA by 1 log		**HR = 2.9 (1.7–4.8); *p* < 0.001**	
Pagliuca, 2019 [82]	PTLD	P and A	Fever at onset of EBV infection (yes vs. no)		**SHR = 6.12 (1.74–21.58); *p* = 0.005**	
Fujimoto, 2019 [55]	PTLD	P and A	Year of HSCT (2010–2015 vs. 1990–2009)		**HR = 1.87 (1.38–2.52); *p* < 0.001**	

Abbreviations: A: adults; Ag: antigen; aGvHD: acute graft-versus-host disease; ALL: acute lymphocytic leukemia; AML: acute myeloid leukemia; ATG: anti-thymocyte globulin; AUC: area under curve; BM: bone marrow; CB: cord blood; CCR5: C–C chemokine receptor 5; cGvHD: chronic graft-versus-host disease; CI: confidence interval; CLL: chronic lymphocytic leukemia; CMV: cytomegalovirus; CsA: cyclosporine A; D+: donor positive; D−: donor negative; D/R: donor/recipient; EBV: Epstein–Barr virus; FFP: fresh-frozen plasma; GvHD: graft-versus-host disease; HL: Hodgkin’s lymphoma; HLA: human leukocyte antigen; HR: hazard ratio; HSCT: hematopoietic stem cell transplantation; IFNG: interferon-γ gene; MAC: myeloablative conditioning; MDS: myelodysplastic syndrome; MMF: mycophenolate mofetil; MMRD: mismatched related donor; MMUD: mismatched unrelated donor; MoAb: monoclonal antibody; MPD: myeloproliferative disease; MRD: matched related donor; MSC: mesenchymal stromal cells; MURD: matched unrelated donor; NHL: non-Hodgkin’s lymphoproliferative disease; NK: natural killer cells; NMAC: non-myeloablative conditioning; NR: not reported; NST: needing systemic therapy; OR: odds ratio; P: pediatric; P and A: pediatric and adult; PB: peripheral blood; PBSC: peripheral blood stem cells; PTLD: post-transplant lymphoproliferative disease; PLT: platelets; R+: recipient positive; R−: recipient negative; RBC: red blood cell; RIC: reduced-intensity conditioning; RR: relative risk; SCT: stem cell transplant; SHR: subhazard ratio; SRBC: sheep red blood cell; TBI: total body irradiation; TCD: T-cell depletion; URD: unrelated donor; vs.: versus. ^╪^ Time-dependent covariate. * Statistically significant associations are shown in bold.

## Data Availability

We confirm that our data are available.

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
