# Peer review of "Factors Associated with Post-Transplant Active Epstein-Barr Virus Infection and Lymphoproliferative Disease in Hematopoietic Stem Cell Transplant Recipients: A Systematic Review and Meta-Analysis"

_vaccines, 2021, doi:10.3390/vaccines9030288_

Round 1

Reviewer 1 Report

The article titled "Factors associated with post-transplant active Epstein-Barr virus infection and lymphoproliferative disease in hematopoietic stem cell transplant recipients: A systematic review and meta-analysis " by Bonong et al. is well-written informative review article. I will strongly recommend the publication with few minor reviews.

1) If possible Author should cite/discuss more works from their one group

2) I would recommend the Author provide a graphical representation or flow chart of the hypothesis clearly showing the correlation between post-transplant EBV and the immune cell components.

Author Response

1) If possible Author should cite/discuss more works from their one group

Response: Thank you to the reviewer for giving us the opportunity to better discuss our own work. We have added two sentences in the Discussion to integrate recent work from our group (see lines 495-498 and lines 515-518 ; modifications with track change).

2) I would recommend the Author provide a graphical representation or flow chart of the hypothesis clearly showing the correlation between post-transplant EBV and the immune cell components.

Response: We again thank the reviewer for this comment. Post-transplant lymphoproliferative disease involves a mechanistically complex multifactorial process, as described in our earlier review article (Tanner JE and Alfieri C. 2001. The Epstein-Barr virus and post-transplant lymphoproliferative disease: interplay of immunosuppression, EBV, and the immune system in immune pathogenesis. Transplant Infect Dis 3: 60-69). It is difficult to provide a simplified graphical representation or flowchart without leaving out important components. Further, seeing that the paper is already quite charged, we opted to simply add 2 phrases to the Introduction section of the paper (see line 52 and lines 56-57).  

Reviewer 2 Report

This is a well written manuscript focused on a systematic review and meta-analysis of factors associated with post-transplant active Epstein-Barr virus infection and lymphoproliferative disease in hematopoietic stem cell transplant recipients. The authors performed an extensive review of the literature covering 74 years using specific inclusion criteria which are well explained in the manuscript. Of the 3362 articles reviewed 77 were analyzed.  The data presented is appropriate and likewise the conclusions drawn by these investigations is solely based upon the data. The investigators also pointed areas of caution concerning interpretation of the data. Overall in my opinion the manuscript is suitable for publication. My only concern is the length of Tables 1 and 2 but I don’t see how these could be shortened without distracting from the overall manuscript.

Author Response

Thank you to the reviewer for the very positive comments. We agree that Tables 1 and 2 cannot be shortened without eliminating important information.

Reviewer 3 Report

I would like to congratulate with the authors. I have made more than 1000 peer review, and it is the first time I am not able to find revisions to be suggested. 

The manuscript has been well presented, data analysis has been conducted rigorously, and the results have been properly detailed with iconographic materials. Discussion section is able to summarize the finding providing readers useful novel insights. 

Author Response

It is an honor to receive such positive comments. Thank you.

Reviewer 4 Report

Well summary the systematic review, and well organized the associated papers, thanks for authors' contributions to the fields in a timely manner.

Author Response

Thank you to the reviewer for the very positive comments.